# Degradation of CP4-EPSPS with a Psychrophilic Bacterium *Stenotrophomonas maltophilia* 780

**DOI:** 10.3390/biom12020318

**Published:** 2022-02-17

**Authors:** Yanhong Peng, Wencong He, Yunjing Li, Lu Liu, Binyang Deng, Guangbo Yan, Jun Yang, Fei Wang, Lixin Ma, Gang Wu, Chao Zhai

**Affiliations:** 1State Key Laboratory of Biocatalysis and Enzyme Engineering, School of Life Sciences, Hubei University, Wuhan 430062, China; pengyanhong666@126.com (Y.P.); hewencong@topgenebio.com (W.H.); lu.liu@innoventbio.com (L.L.); dby0525@outlook.com (B.D.); guangbo_yan@126.com (G.Y.); youcojun@outlook.com (J.Y.); wangfei@hubu.edu.cn (F.W.); malixing@hubu.edu.cn (L.M.); 2Oil Crops Research Institute, Chinese Academy of Agricultural Sciences, Wuhan 430062, China; liyunjing@caas.cn

**Keywords:** CP4-EPSPS, protease, *Stenotrophomonas maltophilia*

## Abstract

CP4-EPSPS (*Agrobacterium* sp. strain CP4 5-enolpyruvylshikimate-3-phosphate synthase) protein showed remarkable thermostability and was highly resistant to proteases, such as trypsin. In order to eliminate the pollution of CP4-EPSPS from the accumulated straws to the surrounding environment during the winter, the present study investigated the extracellular proteases of 21 psychrophilic strains isolated from the south polar region. The results indicated that *Stenotrophomonas maltophilia* 780 was able to degrade CP4-EPSPS at 18 °C efficiently. Further study indicated that it was able to grow in the extract of Roundup Ready soybean at 18 °C, with CP4-EPSPS degraded to an undetectable level within 72 h. The extracellular proteases of *Stenotrophomonas maltophilia* 780 are thermo-sensitive, with an optimal temperature of 65 °C. The genomic sequencing result indicated that this strain had more than a hundred putative protease and peptidase coding genes, which may explain its high capability in decomposing CP4-EPSPS.

## 1. Introduction

GM (Genetically Modified) crops have been planted worldwide over the past decades. During 1996 to 2016, the acreage of GE (Genetically Engineered) crops expanded 109-fold. Among them, the crop covering the most hectares globally is soybean. In 2016, 78% of soybeans planted were GE varieties [1]. There are 30 events of transgenic soybeans, and one of the most representative transgenic soybeans is the Roundup Ready soybean (RRS, event GTS40-3-2). The transgenically expressed protein in RRS is CP4-EPSPS protein, a variant of plant 5-enolpyruvyl-shikimate-3-phosphate synthase (EPSPS). Plant EPSPS is involved in the Shikimate pathway, which plays a critical role in many plant functions, such as the production of aromatic amino acids [2,3]. CP4-EPSPS was originally isolated from the soil *Agrobacterium* sp. strain CP4. The full length of this gene encodes a protein of 55.6 kDa, including a 72 aa chloroplast transit peptide and a mature protein of 47.6 kDa. As a variant of plant EPSPS, it has a lower binding affinity to glyphosate [4]. Therefore, constitutive expression of CP4-EPSPS in crops confers increased tolerance to the Roundup family of agricultural herbicides. Except for soybean [5,6], CP4-EPSPS was also applied in transgenic rice [7], cotton (event MON 1445, MON 88913) [8], corn, etc.

Even though the commercial use of GM crops has increased dramatically, the safety of GM plants is still a controversial topic, attracting both the attention of the scientific community and the public. Therefore, the effects of transgenic plants on domestic animals and the environment have been intensively studied [9]. CP4-EPSPS shows no obvious similarity with proteins associated with allergy and toxicity [10], and several investigations indicated that doses of CP4-EPSPS much higher than the dietary supplement level had no obvious toxicological effect on the reproductive function in rats [11,12] or animal health as a dietary supplementation [13]. However, some research has still raised possibilities that these plants may cause unintended effects via different pathways, which have to be systematically evaluated. Therefore, any genetically modified plants or animals should undergo strict safety evaluation before entering the food market. Previous studies indicated that CP4-EPSPS protein is very stable, and a trace amount of the transgenic protein was still detectable in the highly processed RRS soybean products [14]. Moreover, a higher level of CP4-EPSPS was observed in leaf tissues than in the seeds and roots [10]. To this end, the accumulation of CP4-EPSPS during the returning of straw to the field may cause environmental risks in the long term. In the present study, we screened five isolates with high cold-active protease activity from 21 psychrophilic strains from the south polar region. One of them was able to degrade recombinant CP4-EPSPS efficiently. The molecular taxonomy study indicated that it belonged to *Stenotrophomonas maltophilia*; hence, it was named *S. maltophilia* 780. Further study indicated that *S. maltophilia* 780 was able to grow in the extract of Roundup Ready soybean at 18 °C, with CP4-EPSPS degraded to an undetectable level within 72 h. Accordingly, this isolate showed great potential in eliminating the pollution of the CP4-EPSPS protein to the surrounding environment.

## 2. Materials and Methods

### 2.1. Bacteria, Plasmids, Media, and Reagents

*E. coli* XL10-Gold was obtained from Invitrogen (Carlsbad, CA, USA). *E. coli* BL21 (DE3) were stored in our lab. All psychrophilic bacteria were obtained from the China Center for Type Culture Collection (CCTCC, Wuhan, China). The geographical sources of the strains are listed in Appendix A. Vector pET28a was obtained from Novagen, Merck (Darmstadt, Germany). The ORF of CP4-EPSPS was synthesized by Genecreate (Wuhan, China) and cloned into pET28a directly or fused with the coding sequence of 6 × His to generate N- or C-terminal fusion proteins. The recombinant plasmids were named pET28a-EPSPS, pET28a-His×6-EPSPS, and pET28a-EPSPS-His×6, respectively. Luria–Bertani medium (LB) was prepared as described in the Manual of Molecular Cloning. Kanamycin of 50 µg/mL was supplemented before use. R_2_A media were purchased from Churui (Jiangsu, China), which contained 0.05% yeast extract, 0.05% tryptone, 0.05% casein, 0.05% glucose, 0.05% soluble starch, 0.03% K_2_HPO_4_, 0.0024% MgSO_4_, and 0.03% α-ketopropionic acid sodium. Approximately 1.5% of agar was added for the solid medium. Skimmed milk powder was purchased from BD (USA). Test strips for CP4 EPSPS were obtained from the Institute of Oil Crops, Academy of Agricultural Sciences (Wuhan, China). QualiPlate kit for CP4-EPSPS Corn & Cotton was purchased from Envirologix (Portland, ME, USA). Trypsin was purchased from Biosharp (Guangzhou, China). All other chemicals were analytical reagents.

### 2.2. Expression and Purification of the Recombinant CP4-EPSPS

The recombinant plasmids pET28a-EPSPS, pET28a-His6-EPSPS, and pET28a-EPSPS-His6 were transformed into *E. coli* BL21 (DE3) for the expression of recombinant proteins. The recombinant strains expressing CP4-EPSPS were inoculated into 1 mL of LB media and incubated at 37 °C and 250 rpm overnight. The starter culture was transferred to 50 mL of fresh LB media. The mixture was incubated at 37 °C to OD_600_ of 0.6–0.8, and then induced with 0.5 mM of IPTG at 18 °C for 24 h. Cells were collected and resuspended in lysis buffer (50 mM Tris-HCl, 200 mM NaCl, 50 mM NaH_2_PO_4_, 25 mM imidazole, and 10% glycerol; pH 8.0) with lysozyme to a final concentration of 1 mg/mL.

To purify the target protein bearing the 6 × His tag with Ni-NTA resins, the crude cell lysate was applied to Ni-NTA beads for affinity purification. The column was washed twice with two column volumes of the wash buffer (50 mM Tris-HCl, 200 mM NaCl, 50 mM NaH_2_PO_4_, and 40 mM imidazole; pH 8.0). One column volume of elution buffer (50 mM Tris-HCl, 200 mM NaCl, 50 mM NaH_2_PO_4_, and 200 mM imidazole; pH 8.0) was used to recover the target protein. The sample was then collected and dialyzed with a Millipore 10 kDa cut-off membrane at 4 °C to remove ions, salts, and imidazole, followed by resuspension with storage buffer (50 mm Tris-HCl; pH 7.5).

To purify the target protein without an affinity tag, 300–500 µL of the crude cell lysate was loaded on the native gel. The sample was resolved at 90 v until the loading dye reached the bottom of the gel. A slice of the gel (approximately 2 cm in width) was then cut and stained with the quick staining kit from Beyotime (Shanghai, China). Subsequently, the gel slice containing the target protein was cut from the PAGE gel based on the result of staining and soaked in 300–500 µL of storage buffer for 10–12 h at 4 °C to recover the target protein.

### 2.3. SDS–PAGE Assay of the Proteins of Interest

SDS–PAGE was performed on a 12% running gel. After electrophoresis, the proteins were stained with Coomassie Brilliant Blue R-250. The total protein concentrations of the supernatant were measured using a Bradford kit from Beyotime (Shanghai, China) with bovine serum albumin as the standard.

### 2.4. Analyzing the Stability of Recombinant CP4-EPSPS

To investigate the stability of CP4-EPSPS at room temperature, the recombinant protein was diluted to a final concentration of 2 mg/mL and incubated at room temperature for a month. Approximately 10 µL of the samples was collected every 2 days. Time course monitoring of the stability of CP4-EPSPS to heat was performed with boiling in a water bath or microwaving for more than 2 h.

To study the effect of protease on the stability of CP4-EPSPS, trypsin was added to CP4-EPSPS at a molar ratio of 1:10 and incubated at pH 7.0 for more than 96 h, followed by analysis with SDS–PAGE.

### 2.5. Analyzing the Extracellular Protease Activity of the Psychrophilic Bacteria from the South Polar Region

The psychrophilic bacteria from the polar region were stroked on 1/2R_2_A plates and incubated at 18 °C for 2–3 days, and then a single colony of each strain was inoculated on 1/2R_2_A plates supplemented with 1% skimmed milk. The plates were incubated at 18 °C. If the bacteria secreted extracellular proteases, the milk around the colony would be degraded and a clear halo would appear.

### 2.6. Digestion of CP4-EPSPS with Extracellular Proteases Generated by Psychrophilic Bacteria

The psychrophilic bacteria were incubated in 5 mL of 1/2R_2_A supplemented with 1% milk for 3–4 days. Approximately 200 µL of the culture was centrifuged every 12 h and the extracellular protease activity in the supernatant was tested on 1/2R_2_A plates supplemented with 1% skimmed milk until clear halos appeared. Subsequently, the cell culture was centrifuged at 5000 rpm for 5 min and the supernatant was sterilized with a 0.22 µm syringe filter. The filtrate was mixed with CP4-EPSPS at a molar ratio of 1:10 and incubated at 18 °C. The supernatant incubated at 100 °C for 30 min was used as the negative control. 

### 2.7. Isolating a Pure Culture of S. maltophilia 780 and Microscopy Assay

CCTCC2016780 strain stored in a −80 °C freezer was thawed and a loop of cells was resuspended in 100 µL of 1/2R_2_A media, followed by 10-fold serial dilution with 1/2R_2_A media; then 100 µL of 10^5^, 10^6^, and 10^7^-fold diluted samples were spread on 1/2R_2_A plates and incubated at 18 °C for 2–3 days. The petite colonies (*S. maltophilia* 780) were selected for further experiments. The morphology of the cells was analyzed with Leica microsystems CMS GmbH (Mannheim, Germany) before and after Gram staining. The Gram staining was performed as previously reported. Briefly, a bacterial smear was placed on a slide and then heat-fixed. A drop of crystal violet was added to the sample. Next, the slide was flushed with iodine solution, which fixed the crystal violet to the cell wall, followed by washing with ethanol to remove non-fixed crystal violet. The counterstain safranin was added to color the cells red.

### 2.8. 16S rDNA Sequence Assay and Genomic Sequencing

16S rDNA-based taxonomic analysis and the genomic sequencing of *S. maltophilia* 780 were carried out at Genewiz (Jiangsu, China). The primers used for the amplification of 16S rDNA were 27F: 5′-AGAGTTTGATCCTGGCTCAG-3′ and 1492R: 5′-GGTTACCTTGTTACGACTT-3′. 

### 2.9. Inducing Extracellular Proteases of S. maltophilia 780 with Various Nitrogen Sources

The bacterial strain was suspended in 1 mL of 1/2R_2_A, and then 100 µL of the cell culture was inoculated in 5 mL of 1/2R_2_A supplemented with 1% of various nitrogen sources, i.e., milk, trypton, (NH_4_)_2_SO_4_, yeast extract, and soybean extract. The samples were incubated at 18 °C for 3–4 days and the extracellular protease activity was tested on 1/2R_2_A plates supplemented with 1% skimmed milk.

### 2.10. Quantitatively Analyzing the Extracellular Proteases of S. maltophilia 780

The enzyme activity was measured with casein as a substrate. To study the optimal temperature of the extracellular proteases of *S. maltophilia* 780, 1 mL of a diluted enzyme sample was mixed with an equal volume of 2% (*w*/*v*) casein in 20 mm phosphate buffer (pH 7.5), followed by incubating at a 0 to 80 °C gradient with intervals of 10 °C for 10 min. The reaction was terminated with 2 mL of 10% (*w*/*v*) trichloroacetic acid (TCA). The mixture was centrifuged at 14,000× *g* for 10 min and the OD_280_ of the supernatant was measured to determine the amount of L-tyrosine released during the reaction. To study the thermostability of the extracellular proteases of *S. maltophilia* 780, the diluted enzyme sample was incubated at 25 to 75 °C with intervals of 10 °C for 5–30 min, followed by adding an equal volume of 2% (*w*/*v*) casein in 20 mM phosphate buffer (pH 7.5), followed by incubating at 25 °C for 10 min. As mentioned above, the reaction was terminated with TCA and the OD_280_ was measured. The sample containing the inactivated enzymes (heated at 100 °C for half an hour) and casein was used as the blank to eliminate the background value caused by the L-tyrosin in the casein. A standard curve was plotted using 0 to 60 µg/mL of L-tyrosin.

To obtain the kinetic parameters for the extracellular proteases of *S. maltophilia* 780, a pseudo-one-substrate kinetic model was used as described previously [15]. The kinetic parameters were determined from three independent initial rate measurements performed with the same batch of enzyme. The concentration of casein was 0 to 5.0 µm. The initial rate was measured by the analysis of the produced L-tryosin. The initial rate data were fitted to a Michaelis–Menten equation. The K_M_ and *k*_cat_ values were calculated from the slope and y intercept of the double reciprocal plot drawing with Original 8.0 software of OriginLab (Northampton, MA, USA).

### 2.11. Preparation of Roundup Ready Soybean (RRS) Extract

RRS was ground and 5 g of the powder was added to 250 mL of dH_2_O, followed by vigorous stirring for 30 min. The sample was centrifuged at 5000 rpm for 5 min. The supernatant was sterilized by filtration using a 0.22 µm filter (Millipore, MA, USA).

### 2.12. ELISA

ELISA was carried out as described in the Manual of QualiPlate kit for CP4-EPSPS Corn & Cotton (Envirologix, ME, USA). Briefly, 50 µL of Roundup Ready enzyme conjugate and 50 µL of the sample were added to each well of the plate. The samples were mixed thoroughly, followed by incubating at room temperature for 45 min. The solution in the wells was removed by vigorous shaking and the wells were washed with the wash buffer three times. Equal volumes of the substrate were added to each well and the plate was incubated at room temperature for 15 min, followed by adding 100 µL of the stop solution to each well. The result was read with a multi-mode microplate reader (SpectraMax, CA, USA) at a wavelength of 450 nm.

## 3. Results

### 3.1. Recombinant Expression of CP4-EPSPS

*E. coli* BL21 (DE3) bearing pET28a-EPSPS, pET28a-His6-EPSPS, and pET28a-EPSPS-His6 were used to express the recombinant proteins. The ORF of CP4-EPSPS is 1623 bp (Genbank accession No. NP_740524, Appendix A) and encodes a protein of approximately 47 kDa. The result of SDS–PAGE indicated that a band consistent with the predicted size of CP4-EPSPS was detected in the supernatant of the cell lysate (Figure 1a). The target proteins bearing 6 × His tags were purified with Ni-NTA. Meanwhile, the recombinant CP4-EPSPS without a fusion tag was purified with native gel. The result of SDS–PAGE indicated that all of them were purified successfully (Figure 1b–d).

### 3.2. Stability of the Recombinant CP4-EPSPS

To investigate the stability of CP4-EPSPS, the recombinant protein was diluted to 2 mg/mL and incubated at room temperature. The result of SDS–PAGE indicated that no obvious degradation of the target protein was detected in 25 days (Figure 2a). The three forms of recombinant proteins were also boiled in a water bath and no obvious degradation of the target protein was detected after 1.5 h of heating (Figure 2b). In addition, the target protein remained soluble after 30 min of microwaving (Figure 2c). These results indicated that CP4-EPSPS was highly resistant to heat. To investigate the effect of protease on CP4-EPSPS, commercial trypsin was used to digest CP4-EPSPS. The result indicated that CP4-EPSPS was resistant to high concentrations of trypsin. It took 96 h for trypsin to degrade CP4-EPSPS completely at a molar ratio of 1:10 (Figure 3).

### 3.3. Screening Psychrophilic Bacteria Strains Capable of Degrading the Recombinant CP4-EPSPS

Twenty-one psychrophilic bacterial strains isolated from the south polar region (Appendix A) were inoculated on ½R_2_A plates supplemented with 1% milk and incubated at 18 °C. Clear halos appeared around the colonies of CCTCC2016780, 741, 742, 743, and 808, while a relatively small halo formed around CCTCC2016809 after 3 days (Figure 4), which indicated that these strains secreted cold-active proteases to the surrounding environment. The experiment was run in triplicate. CCTCC2016780, 741, 742, 743, and 808 exhibited robust extracellular protease activities and clear halos were formed in all three repeats. These strains were incubated in 1/2R_2_A supplemented with 1% milk for 3 days. The supernatants of the cell cultures were gathered and incubated with the recombinant CP4-EPSPS. After 48 h of digestion, proteases secreted by CCTCC2016780 and 808 showed obvious degradation of CP4-EPSPS and the band of CP4-EPSPS almost disappeared. The extracellular proteases of CCTCC2016741 and 742 were also capable of degrading CP4-EPSPS and the target protein was partially digested. On the contrary, CCTCC2016809 exhibited no obvious effect on CP4-EPSPS (Figure 5). Since CCTCC2016780 demonstrated the most obvious effect, it was used for further study.

### 3.4. Identification of Psychrophilic Bacteria CCTCC2016780

According to the molecular taxonomy result of CCTCC, CCTCC2016780 belongs to *Mucilaginibacter gotjawali* (Appendix A). During the process of obtaining a pure culture, we realized that this isolate was composed of two strains. Both strains formed white colonies on 1/2R_2_A plates after 48 h of incubation at 18 °C (Figure 6). Both strains were identified by 16S rDNA sequence analysis. The strain that showed robust growth was identified as *Pseudomonas* sp., since its 16S rDNA had 100% identity with *Pseudomonas moraviensis* WTB8 and *Pseudomonas* sp. PS13 (Appendix A). The 16S rDNA of the petite colonies showed 99% identity with *Stenotrophomonas maltophilia* 7K14 and R551-3, which suggested that it belonged to *Stenotrophomonas maltophilia* (Appendix A). Accordingly, these two strains were named *Pseudomonas* sp. 780 and *S. maltophilia* 780, respectively.

The extracellular proteases of both strains were induced with milk and utilized for the degradation of the recombinant CP4-EPSPS. The result of SDS–PAGE indicated that the extracellular proteases of *S. maltophilia* 780 degraded CP4-EPSPS efficiently, and almost all CP4-EPSPS was degraded in 5 h (Figure 7). On the contrary, the extracellular proteases of *P*. sp. 780 had no effect on the target protein (Appendix A). Therefore, *S. maltophilia* 780 was used for further study.

The morphological study indicated that *S. maltophilia* 780 was Gram negative and had a short rod shape (Figure 8), which is consistent with the typical features of *S. maltophilia*. As a psychrophilic bacterium, *S. maltophilia* 780 showed obvious growth at 18 °C, while its growth was completely inhibited at 37 °C in both 1/2R_2_A and LB media.

### 3.5. Characteristics of the Extracellular Proteases of S. maltophilia 780

The effect of various nitrogen sources on the biosynthesis of extracellular proteases of *S. maltophilia* 780 was investigated. The organic nitrogen sources had stronger inducing effects than simple nitrogen sources, such as (NH_4_)_2_SO_4_. Among the organic nitrogen sources, milk demonstrated the most obvious effect (Figure 9). Because *S. maltophilia* 780 is a psychrophilic bacteria, its extracellular proteases had weak activity at 0 °C and the activity increased with the elevation of the temperature (Figure 10a). It is puzzling that the activity reached the maximum level at 65 °C, which was relatively high in comparison with other psychrophiles. Consistent with the features of cold-active enzymes, the extracellular proteases of *S. maltophilia* 780 were sensitive to heat and most of the activity was lost at 70 °C in 10 min. On the other hand, they were relatively stable at 25 °C and retained approximately 80% of the activity after 30 min (Figure 10b). The kinetic parameters of the extracellular proteases of *S. maltophilia* 780 were investigated. The binding affinity (*K*_M_) was 767.08 µm. On the contrary, the catalytic turnover (*k_cat_*) was 1.39 × 10^3^ s^−1^ (Figure 10c).

### 3.6. Degradation of CP4-EPSPS in RRS with S. maltophilia 780a

*S. maltophilia* 780 was incubated in 1/2R_2_A supplemented with 1% milk for 3 days, and the supernatant was applied to degrade CP4-EPSPS in RRS extract. The result of the test strips indicated that the concentration of CP4-EPSPS decreased to an undetectable level after 30 h of incubation (Figure 11). As the culture of *S. maltophilia* 780 could degrade CP4-EPSPS in the transgenic soybean efficiently, this isolate was cultivated in the extract of RRS directly. The result indicated that *S. maltophilia* 780 was able to grow in the extract of RRS, while *E. coli* DH5α grew much slower (Figure 12a). Meanwhile, ELISA indicated that CP4-EPSPS in the extract was degraded to an almost undetectable level after approximately 72 h. On the other hand, *E. coli* DH5α had no obvious effect (Figure 12b).

### 3.7. Genomic Sequencing of S. maltophilia 780 and the Genes Encoding Proteases

The general genomic features of *S. maltophilia* 780 are listed in Table 1. The G + C content of *S. maltophilia* 780 was 64.58%, which was consistent with that of other *S. maltophilia* genomes. The genome size of *S. maltophilia* 780, slightly larger than the previously reported *S. maltophilia* genomes, was approximately 5.77 Mb with 5536 predicated CDS [16]. Functional annotation indicated that more than a hundred genes coded proteases and peptidases (Appendix A).

## 4. Discussions

Psychrophiles are defined as organisms having an optimal temperature for growth of 15 °C or lower and a maximum temperature for growth of about 20 °C [17]. Psychrophiles thrive in the cold environments on Earth and provide a rich source of cold-active enzymes, which display high catalytic activity at low temperatures in comparison with their mesophilic counterparts. These enzymes play important roles in psychrophiles to cope with low temperatures [18]. Recently, many psychrophiles have been screened and applied in bioremediation [19,20], and their cold-active enzymes, such as lipase, amylase, cellulase, protease, etc., have also been widely applied in the pharmaceutical, food, and detergent industries [21,22]. Previous reports and the present study indicated that CP4-EPSPS is remarkably stable. After GM crops, such as soybean, rice, etc., are harvested, the straw is returned to the field. During the decomposition of the straw in the autumn and winter, CP4-EPSPS accumulates in the soil. To accelerate the degradation of CP4-EPSPS in soil under low temperatures, we screened bacteria with high cold-active proteases from psychrophilic bacteria. Among them, *S. maltophilia* 780 showed the most obvious effect on decomposing CP4-EPSPS. Previous studies on *S. maltophilia* focused mainly on the comparative genomics [23] and the mechanisms of antibiotic resistance, because some *S. maltophilia* isolated from hospital settings are emerging nosocomial pathogens with multi-drug resistance [24]. However, *S. maltophilia* have been widely discovered in various environments, including water, soil, and especially the rhizosphere. *S. maltophilia* in the rhizosphere promote the growth of plants and protect them against fungal and bacterial pathogens [25]. Some *S. maltophilia* also contribute to bioremediation and phytoremediation [26,27]. Therefore, they display great potential for agricultural and industrial applications. In the present study, we discovered that the extracellular enzymes synthesized by *S. maltophilia* 780 are able to degrade CP4-EPSPS efficiently. Meanwhile, as an Antarctic psychrophile, *S. maltophilia* 780 grows at 18 °C, but is inhibited at 37 °C; hence it might be safe for human health, but this needs further investigation. Therefore, it is possible to supply this isolate to composite microbial fertilizer to eliminate the pollution of CP4-EPSPS in the soil during the autumn and winter; in turn, the robust protease activity of this isolate could convert proteins and peptides in the soil to small molecules and promote the growth of crops in the next year.

The proteases derived from Antarctic organisms have attracted worldwide attention due to their high catalytic activity at low temperatures [15,28,29]. Similar to these cold-active proteases, the extracellular proteases of *S. maltophilia* 780 are thermo-sensitive [30,31,32]. The enzymes lost more than 50% of their activity at 45–75 °C for 30 min. On the other hand, the proteases produced by psychrophilic bacteria are not fully adapted to work in cold environments. The activity of their proteases is temperature dependent, which is similar to their mesophilic counterparts. Their proteolytic activity is almost negligible at the temperature of the environment where the producing strains live. Psychrophilic bacteria turn to produce more enzymes to make up for the loss of activity at low temperatures, rather than increase the activity of the enzymes under low temperatures. Therefore, it is common that the optimal temperatures of proteases reach 40–55 °C [28]. However, the extracellular proteases of *S. maltophilia* 780 exhibited an optimal temperature of 65 °C, which is higher than that in previous reports. We deduced that there were multiple proteases in *S. maltophilia* 780. Consistent with this result, the gene annotation of *S. maltophilia* 780 indicated that there are more than a hundred putative proteases and peptidases in *S. maltophilia* 780, and more proteases may be found in the hypothetical proteins, which rend the gene mining of the proteases responding to the decomposition of CP4-EPSPS a huge challenge. Moreover, the degradation of CP4-EPSPS may be caused by the cooperation of several proteases. For this reason, certain proteases that catalyze the degradation of CP4-EPSPS were not detected in the present study. 

In summary, we screened a psychrophile, *S. maltophilia* 780, for the efficient degradation of CP4-EPSPS, which may provide a powerful tool to solve the pollution of transgenic proteins in the natural environment.

## Figures and Tables

**Figure 1 biomolecules-12-00318-f001:**
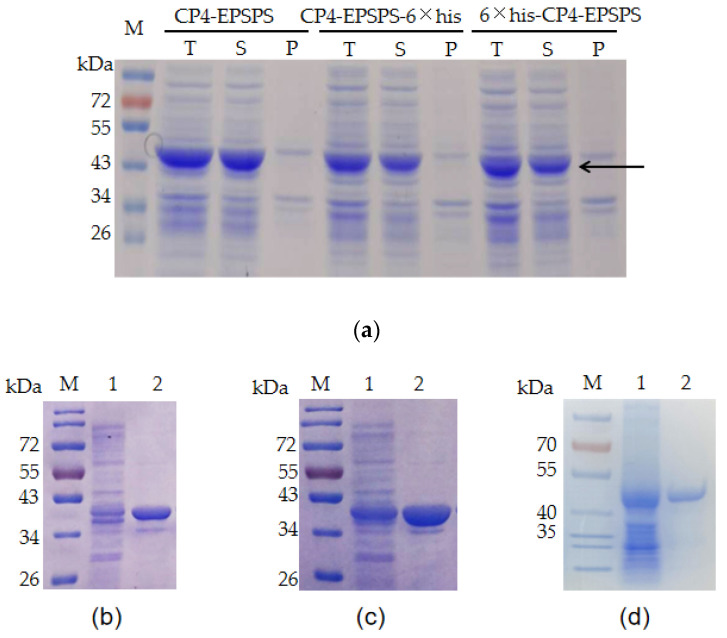
Expression and purification of recombinant CP4-EPSPS protein. (**a**) Expression of recombinant CP4-EPSPS with *E. coli* BL21 (DE3) as a host. T: total protein of the cells; S: supernatant of the cell lysate; P: pellet of the cell lysate. The target protein is indicated with an arrow. (**b**) Purified 6 × his-CP4-EPSPS; (**c**) purified CP4-EPSPS-6 × his; (**d**) purified CP4-EPSPS. Lane 1: supernatant of the cell lysate; Lane 2: purified target protein; M: protein molecular weight marker (the size of each band is indicated on the left).

**Figure 2 biomolecules-12-00318-f002:**
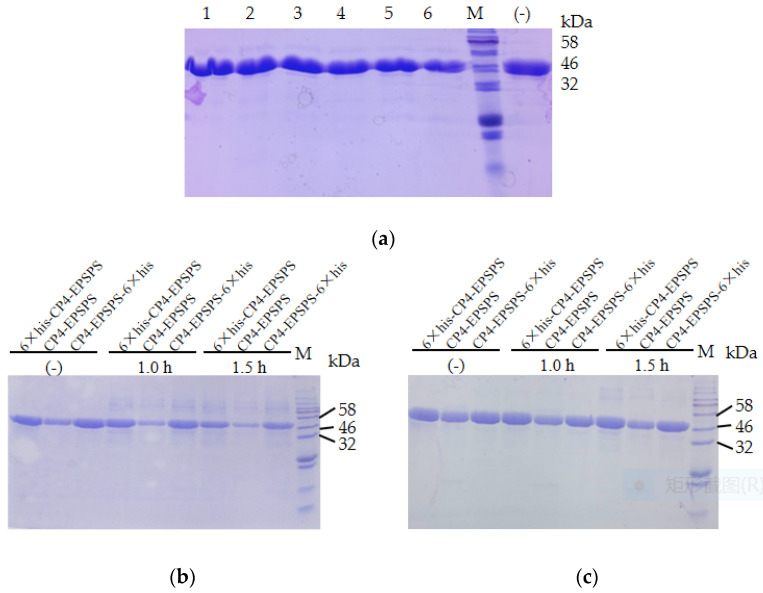
Stability of the recombinant CP4-EPSPS. (**a**) Stability of the recombinant CP4-EPSPS at room temperature. Lane 1–6: recombinant 6 × his-CP4-EPSPS incubated at room temperature for 1, 2, 4, 7, 15, and 25 days, respectively; (−) the negative control. (**b**) Stability of the recombinant CP4-EPSPS with boiling. (**c**) Stability of the recombinant CP4-EPSPS with microwaving; M, protein molecular weight marker (the size of each band is indicated on the right).

**Figure 3 biomolecules-12-00318-f003:**
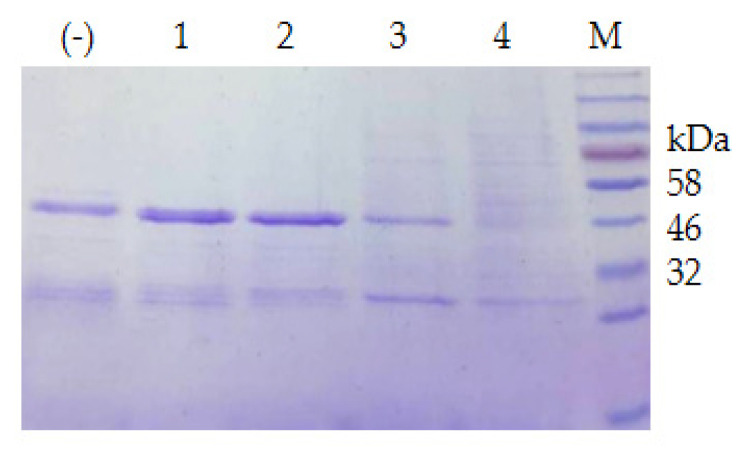
Digestion of CP4-EPSPS with trypsin. Lane 1–4: digestion of 6 × his-CP4-EPSPS with trypsin at a molar ratio of 10:1 for 12, 24, 84, and 96 h, respectively; (−) the negative control; M, protein molecular weight marker (the size of each band is indicated on the right).

**Figure 4 biomolecules-12-00318-f004:**
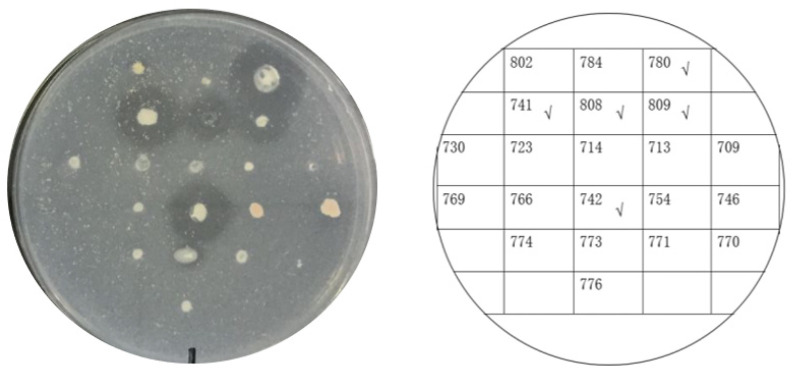
Screening psychrophilic bacterial strains with extracellular protease activity. The serial number of each strain is indicated on the right. Strains with obvious extracellular protease activity are ticked.

**Figure 5 biomolecules-12-00318-f005:**
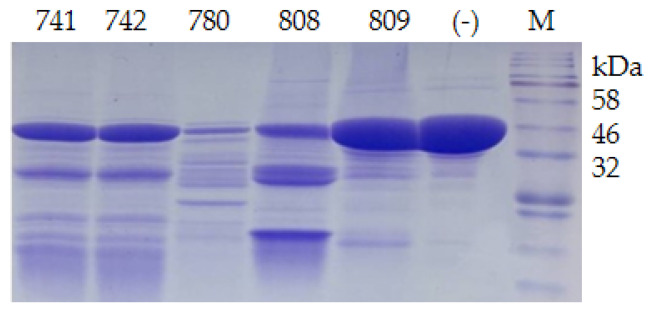
Degradation of recombinant CP4-EPSPS with extracellular proteases of psychrophilic bacterial strains. The serial number of each strain is indicated on the SDS–PAGE gel; (−) the negative control; M, protein molecular weight marker (the size of each band is indicated on the right).

**Figure 6 biomolecules-12-00318-f006:**
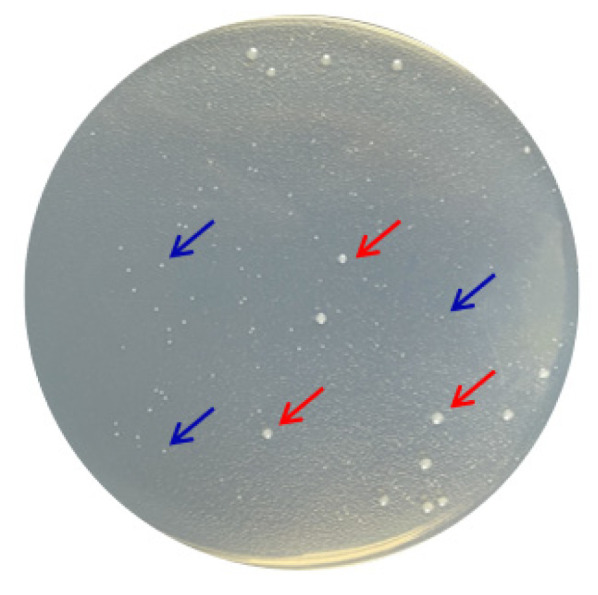
Two bacterial strains isolated from *Mucilaginibacter gotjawali* 780. The red arrows indicate *Pseudomonas* sp. 780 with robust growth and the blue arrows indicate *S. maltophilia* 780, which formed petite colonies.

**Figure 7 biomolecules-12-00318-f007:**
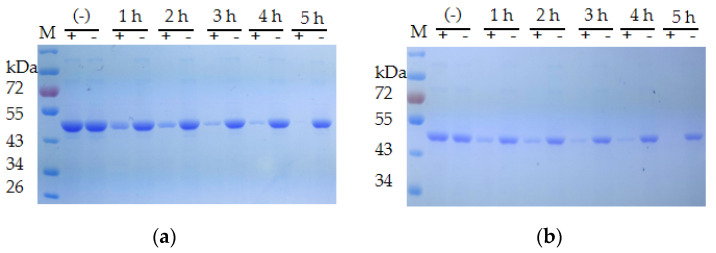
Time course of the degradation of recombinant CP4-EPSPS with the extracellular proteases of *S. maltophilia* 780. (**a**) Degradation of 6 × HIS-CP4-EPSPS; (**b**) degradation of CP4-EPSPS-6×HIS; (−) the negative control (degradation with supernatant of cell culture heated at 100 °C for 30 min); M, protein molecular weight marker (the size of each band is indicated on the left).

**Figure 8 biomolecules-12-00318-f008:**
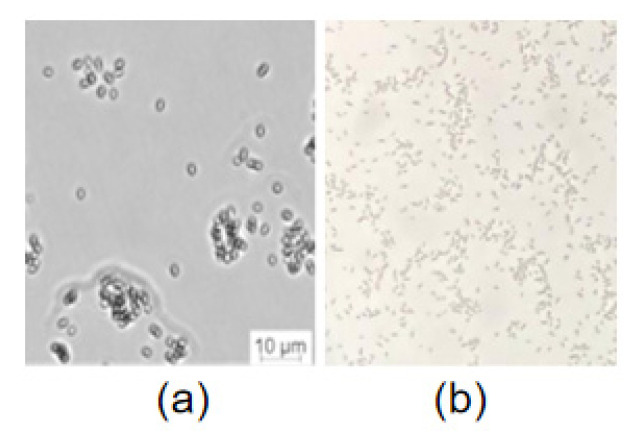
Microscopic examination of *S. maltophilia* 780. (**a**) DIC (digital image correlation) with Leica microsystems CMS GmbH; (**b**) Gram staining.

**Figure 9 biomolecules-12-00318-f009:**
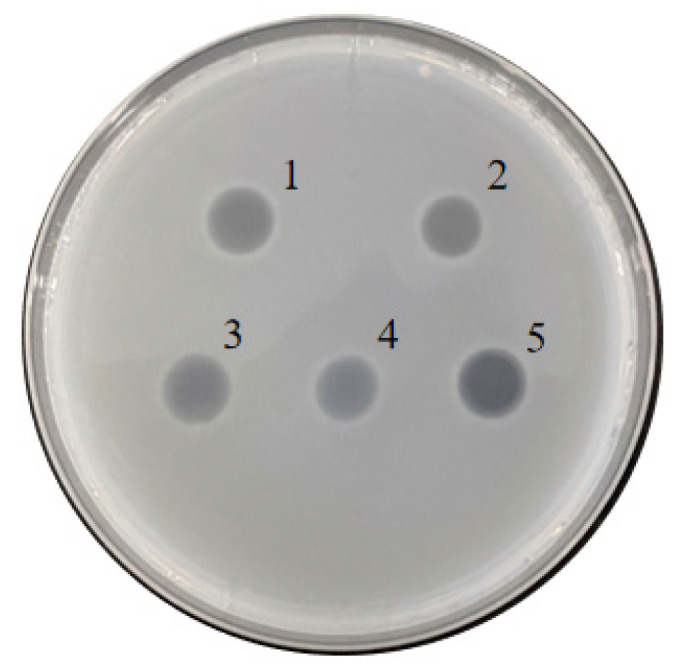
Extracellular protease generated by *S. maltophilia* 780 induced with different nitrogen sources. 1: extract of Roundup Ready soybean powder; 2: 1% yeast extract; 3: 1% trypton; 4: 1% (NH_4_)_2_SO_4_; 5: 1% milk powder.

**Figure 10 biomolecules-12-00318-f010:**
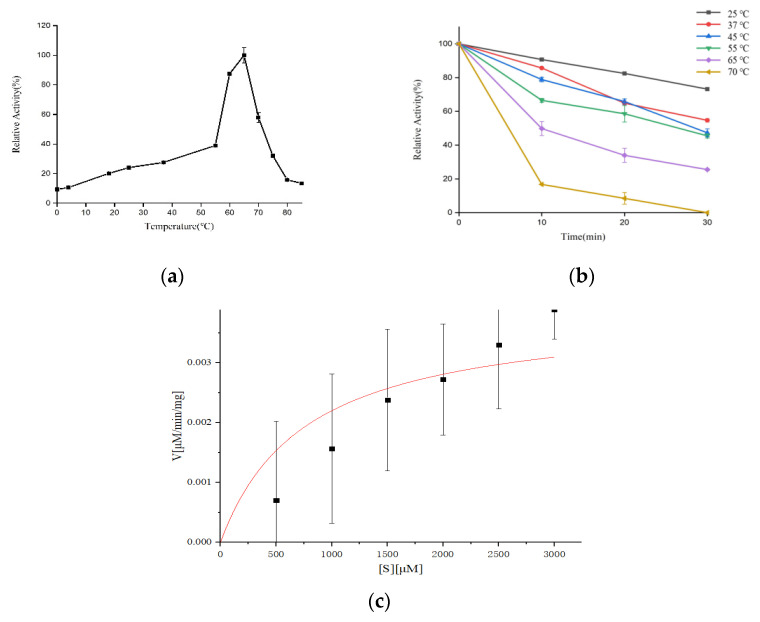
Effect of temperature on the activity and stability of the extracellular proteases of *S. maltophilia* 780. (**a**) Optimum temperature of the enzymes; (**b**) effect of temperature on the thermostability of the enzymes; (**c**) kinetic analysis of the extracellular proteases of *S. maltophilia* 780.

**Figure 11 biomolecules-12-00318-f011:**
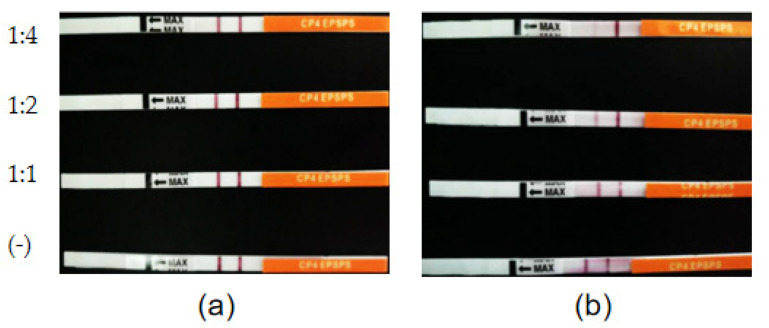
Time course of the degradation of CP4-EPSPS in the extract of RRS using the cell culture of *S. maltophilia* 780. (**a**) RRS extract incubated with inactivated enzymes (negative controls). (**b**) RRS extract incubated with the enzymes. The ratio of the extract to the enzymes (*v*/*v*) is indicated on the left of the figure.

**Figure 12 biomolecules-12-00318-f012:**
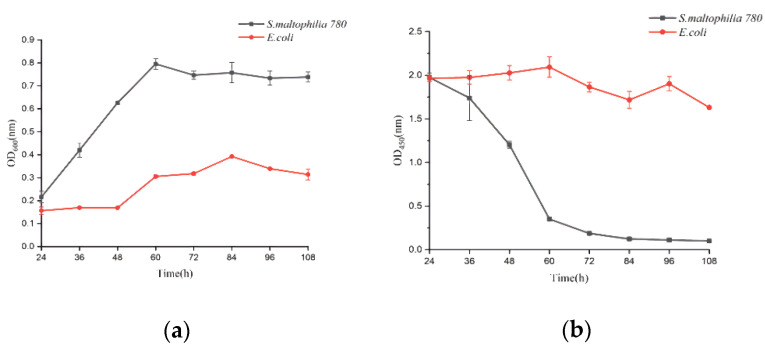
Degradation of CP4-EPSPS in the extract of RRS during the growth of *S. maltophilia* 780. (**a**) Growth curve of *E. coli* and *S. maltophilia* 780 in the extract of RRS. (**b**) CP4-EPSPS level in the cell culture.

**Table 1 biomolecules-12-00318-t001:** Genomic features of *S. maltophilia* 780.

Feature	Value
Genome size/Mb	5.77
G + C content/%	64.58%
Protein-coding genes (CDS)	5536
rRNA (5S, 16S, 23S)	3
tRNA	55
Other ncRNA	54

## Data Availability

Not applicable.

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
