# Peer review of "Degradation of CP4-EPSPS with a Psychrophilic Bacterium Stenotrophomonas maltophilia 780"

_biomolecules, 2022, doi:10.3390/biom12020318_

Round 1

Reviewer 1 Report

Review of "Degradation of CP4-EPSPS with a psychrophilic bacterium  Stenotrophomonas maltophilia 780", by Peng etal.

Here the authors have isolated psychrophilic bacteria with the ability to degrade CP4-EPSPS, a protein that its gene is encoded in transgenic crops to enable the used of Roundup as herbicide. The need to degrade residual CP4-EPSPS with increased stability could be of interest to the public.

Major comments:

  1. It would be informative to include in the introduction information on other known bacteria or enzymes with the ability to degrade CP4-EPSPS. Particularly since in the study no specific proteases were identified and characterized, a comparison to the data of CP4-EPSPS degradation with other bacteria would support the state in the abstract "… which may explain its high capability in decomposing CP4-EPSPS" and in the discussion: line 316.
  2. Figure 10 and the description in 3.5. "Characteristics of extracellular proteases of maltophilia 780", is lacking details on how the activity was performed, which assay, concentration of substrate, volume of bacteria, or supernatants. Was it detected in 270nm for tyrosin release? If so, it would be important to show the background value of non-degraded casain in 270nm (same reaction conditions of the substrate alone in buffer, this will show the background activity/control for the enzymatic activity), if this was not performed, then it should be added.
  3. Line 290: " Functional annotation indicated that more than a hundred genes coded proteases and 290 peptidases."-can the author present this result in a table or graph? does the number 290 is high or low compared to other bacteria? psychrophiles bacteria?

      Can the author also comment on the fact that they found more than 100 genes of putative        genes encoding proteases, yet the optimal temperature pick is very narrow? If this was the residual activity of a mixture of enzymes I would suspect a broad range of temperature optimum.

Minor comments:

  1. In line 12: How many strains were screened? lanes 11-14 is very long and not clear, please rewrite for more clarity.
  2. Line 16: "a optimal temperature of" replace with " an optimal temperature of"
  3. Line 16: if its optimum is 65 degrees which is high, why is it considered thermos-sensitive? Please clarify.
  4. Line 17: please check for plural writing.
  5. Many abbreviations in the introduction are not explained in the first time of appearance. GE, GM?
  6. Line 24: "acreage" means? Average?
  7. Line 27; " event 40-3-2”, please clarify numbers
  8. Line 33: replace "less" with lower
  9. Line 76: These are not strains please replace with "E. coli cells" or "bacteria".
  10. Lanes 76-77- was any antibiotics was added to the media?
  11. Line 83: "resin"-replace with resins
  12. Line 89: " to remove ions and salts"- add "and imidazole"
  13. Line 94: " subsequently"-should be capital "S"
  14. Line 107: " To study the effect of protease the stability of CP4-EPSPS" add on after protease. How the stability was analyzed? Via SDS-PAGE? By plates with milk, as described in secession 2.6?? please explain here.
  15. In Material and Method secession 2.5 there is no description corresponding to the title "Analyzing of the extracellular protease activity of the psychrophilic bacteria from south polar 109 region", only growing of the strains, please elaborate on how the extracellular protease activity was analyzed-was it performed as described in 2.4, or 2.6? if so it should be combined, or change the title.
  16. What is the content of media 1/2R2A? should be specified or give a reference.
  17. Lines 117, its not clear what was performed first, was the plating performed before centrifugation? Or the same cultures were used both for plating and centrifugation?
  18. Line 119; what is considered "appropriate ratio"? please explain the measures to determine the ratio, or specify the ratio/range of ratios.
  19. Secession 2.7-not clear why the title is "Isolating pure culture of Mucilaginibacter gotjawali 780", as the isolation part is not described here, only the growing the cells in solid culture after they were stored at -80â—¦
  20. The strains isolation from the environment should be described in more details.
  21. Method in 2.6 should be elaborated, which primers were used for the 16sRNA analysis, which sequencing methods and other relevant details.
  22. In 2.10. Analysis of the protease activity" a specified title should be writing to differentiate from the previous analysis of protease activity in plates supplemented with milk.
  23. Line 140: "appropriate temperature" specify temperatures.
  24. In 2.11. "Preparation of RRS extract" -specify RRS
  25. Line 152 "The samples was" change to were.
  26. Line 153: " wash buffer" describe its content.
  27. In Figure 1; which fraction does Total protein represents? Is it the cells before lysis? Were the volumes of loaded samples from total, supernatant and pellet were the same? no arrow in the figure indicating the bend, as writing in the legend.
  28. In the results in 3.1, the details on the expression vectors and the expression should be briefly described.
  29. Line 201: "The strains showed robust extracellular protease activity", can the authors elaborate how did they detected robust strains-how many time was this assay performed, were there triplicates?. Did they re-grow the selected strains, and verified isolated strains by streaking on agar plates, and performed the assay again to verify the activity? The isolation part was not was not clear.
  30. Lines 211-212, should be re-written for clarity and combined with the previous paragraph.
  31. Description of Microscopy analysis (Figure 8) is missing in MM.
  32. Figure 8: abbreviations in the legend of figure 8 are not explained: DIC, CMS, GmbH.
  33. Line 255, " It is puzzling that the activity reached maximum at 65℃, which was relatively high in comparison with other psychrophiles." This should appear in the discussion.
  34. The Y axis in Figure 10b should be % residual activity and not relative activity.
  35. Line 313: change "promoter" to "promote".
  36. Line 318: "Meanwhile, as an antartic psychrophile, maltophilia 780 318 grows at 18℃ but is inhibited at 37℃, hence it is safe to human health." This last part should be changed to "should/might (or any other relevant word) be safe to human health, but this needs further investigations", as its extracellular proteases will persist at 37%, and their safety should be tested in the future for such application.
  37. The optimal temperature of the enzymatic activity was observed here at 65â—¦ degrees, although this is a psychrophilic bacterium. In this line, a comparison of optimal temperatures of other extra-cellular enzyme from psychrophiles bacteria would be insightful in the discussion. For example, how common is this phenomena? Specifically in proteases, is it related to the physiological role of these enzymes?

  1. I am not sure what is the journal policy regarding data not shown, but I believe that "the extracellular proteases of P. sp. 231 780 had no effect to the target protein" should be presented (maybe in the supporting information).

Author Response

Thank you very much for your suggestions,we have answered all of your questions  and revied the manuscript according to your suggestions,we hope your are satisfied with the revised manuscript.

Reviewer 2 Report

This manuscript entitled "Degradation of CP4-EPSPS with a psychrophilic bacterium Stenotrophomonas maltophilia 780" reports novel data and it can be published in the journal biomolecules.

  Nevertheless, I would strongly suggest to authors to thorougly reconsider their practice as far as it conserns the experimentation on enzyme kinetics . In more details, and by taking as exmples figures 10 (A & B) the "Relative Activity %" is strictly unacceptable among enzymologists, as it is meaningless. Authors should realize that the only acceptable entities, which could deliver substantially information on the kinetic behavior of any enzyme, are the kcat/Km (or Vmax/Km), kcat (or Vmax), and Km. These latter entities, as it is well known, correspond to [S]<<Km, to [S]>>Km, and (in the simplest case) to a measure of affinity between enzyme and its substrate.

Moreover, experimenter may gain much more information on the thermodynamics of the used enzymatic system if as abscissa of the profile is the absolute temperature scale.

 The above comments should be taken into serious account by the authors in their future works.

Author Response

(The authors gave the same response as above.)

Reviewer 3 Report

Comments

The manuscript of Yanhong Peng et al is dedicated to investigation the possibilities of using S. maltophilia 780 for efficient degradation of CP4-EPSPS. This can provide a powerful tool to solve the problem of pollution of natural environment by transgenic proteins. One of the screened isolates from 21 psychrophilic strains from south polar region was able to degrade recombinant CP4-EPSPS efficiently at 18 oC.

This isolate showed a great potential in eliminating the pollution of surrounding environment by CP4-EPSPS protein.

The manuscript may be of interest to a wide range of readers.

I can recommend this manuscript for publication.

One question:

Have experiments been carried out to determine the hydrolysis products of CP4-EPSPS using  electrophoresis, Western blot, or LC-MS?

Minor points

-Line 25:  “GE varieties[1]” – Insert space

-Line 31:  Strain – strain

-Line 36:  rice[7] – Insert space

-Line 40:  studied[9]. – Insert space

-Line 44:  -tion[13]. – Insert space

-Line 49-50:  [..]– Insert space before [

-Line 87: pH8.0 – Insert space

-Line 90, 108 and 140: pH7.5 – Insert space

-Line 145: dH2O  - dist.?

-Line 207, 235, 253, 255, 271, 274, 276, 282:   Fig.No – Insert a space before figure number

- Line 300: tools. – tools). - there must be a parenthesis somewhere

- Line 302, 304: etc – etc.

- Line 310, 312, 328: Insert space before [

-Line 335:   “CP4-EPSPS haven’t” - degradation of CP4-EPSPS was not detected…

-Line 337:  ....a powerful...

Author Response

(The authors gave the same response as above.)

Round 2

Reviewer 1 Report

The authors have addressed all comments, I have few comments regarding the new presented data:

  1. Line 178: "a pseudo-one-substrate kinetic model was used as described previously."-please cite.
  2. Km should be KM
  3. 178: please specify the program was used for the MM fitting
  4. Figure 10 c:  1)  units should be in "[]"nor "()". 2) Y axis is truncated and there are values with a minus sign, this should be corrected. 3) the legend of "c"- the kinetic analysis or MM is presented and not the kinetic parameters.

Author Response

Thank you very much for your suggestions,we hope your are satisfied with the revised manuscript.
